# Two Molecular Plasma-Based Diagnostic Methods to Evaluate Early Infection of *Schistosoma japonicum* and Schistosomiasis Japonica

**DOI:** 10.3390/microorganisms11041059

**Published:** 2023-04-18

**Authors:** Yang Hong, Qinghong Guo, Xue Zhou, Liying Tang, Cheng Chen, Zheng Shang, Kerou Zhou, Zhizhong Zhang, Jinming Liu, Jiaojiao Lin, Bin Xu, Jun-Hu Chen, Zhiqiang Fu, Wei Hu

**Affiliations:** 1National Institute of Parasitic Diseases, Chinese Center for Diseases Control and Prevention (Chinese Center for Tropical Diseases Research), Key Laboratory of Parasite and Vector Biology, National Health Commission of the People’s Republic of China (NHC), World Health Organization (WHO) Collaborating Center for Tropical Diseases, National Center for International Research on Tropical Diseases, Shanghai 200025, China; 2National Reference Laboratory for Animal Schistosomiasis, Key Laboratory of Animal Parasitology of Ministry of Agriculture and Rural Affairs, Shanghai Veterinary Research Institute, Chinese Academy of Agricultural Sciences, Shanghai 200241, Chinafuzhiqiang@shvri.ac.cn (Z.F.); 3Laboratory of Environmental Entomology, College of Life Sciences, Shanghai Normal University, Shanghai 200234, China; 4School of Basic Medical Sciences and Forensic Medicine, Hangzhou Medical College, Hangzhou 310013, China; 5State Key Laboratory of Genetic Engineering, Ministry of Education Key Laboratory of Contemporary Anthropology, Department of Microbiology and Microbial Engineering, School of Life Sciences, Fudan University, Shanghai 200438, China

**Keywords:** *Schistosoma japonicum*, qPCR, RPA–LFD, early infection, diagnosis

## Abstract

The prevalence and infectious intensity of schistosomiasis japonica has decreased significantly in China in the past few decades. However, more accurate and sensitive diagnostic methods are urgently required for the further control, surveillance, and final elimination of the disease. In this study, we assessed the diagnostic efficacy of a real-time fluorescence quantitative PCR (qPCR) method and recombinase polymerase amplification (RPA) combined with a lateral-flow dipstick (LFD) assay for detecting early infections of *Schistosoma japonicum* and different infection intensities. The sensitivity of the qPCR at 40 days post-infection (dpi) was 100% (8/8) in mice infected with 40 cercariae, which was higher than in mice infected with 10 cercariae (90%, 9/10) or five cercariae (77.8%, 7/9). The results of the RPA–LFD assays were similar, with sensitivities of 55.6% (5/9), 80% (8/10), and 100% (8/8) in mice infected with 5, 10, and 40 cercariae, respectively. In goats, both the qPCR and RPA–LFD assays showed 100% (8/8) sensitivity at 56 dpi. In the early detection of *S. japonicum* infection in mice and goats with qPCR, the first peak in positivity appeared at 3–4 dpi, when the positivity rate exceeded 40%, even in the low infection, intensity mice. In the RPA–LFD assays, positive results first peaked at 4–5 dpi in the mice, and the positivity rate was 37.5% in the goats at 1 dpi. In conclusion, neither of the molecular methods produced exceptional results for the early diagnosis of *S. japonicum* infection. However, they were useful methods for the regular diagnosis of schistosomiasis in mice and goats.

## 1. Introduction

Schistosomiasis is a severe zoonotic parasitic disease caused by three principal species of the blood fluke *Schistosoma* in humans, and is distributed in 78 tropical and subtropical countries and areas [1]. In China, only *S. japonicum* is epidemic, and is mainly distributed along the Yangtze River, causing schistosomiasis japonica. Over the past few decades, there have been remarkable advances in the prevention and control of schistosomiasis in China. Today, schistosomiasis japonica has been eliminated from many previously endemic regions, and its prevalence in humans and livestock in most regions of China shows both a low incidence rate and low infection intensity. In 2020, only three cases in human (*n* = 273,712) and none in domestic animals (*n* = 130,673) were detected nationwide [2].

Although the currently available parasitological and immunological diagnostic assays have been widely used in the national schistosomiasis control program in China, such as the Kato–Katz (KK) technique, miracidium hatching test (MHT), indirect hemagglutination assay (IHA), and enzyme-linked immunosorbent assay (ELISA), these methods have several shortcomings and face challenges in the current situation. Parasitological diagnosis has been the “gold standard” until now, and is suitable for field testing because it is simple to perform and is inexpensive. The KK technique is recommended by the World Health Organization to confirm intestinal schistosomiasis infection [3]. MHT is more widely used for animal schistosomiasis diagnosis in China. However, the procedure is usually time-consuming, requires trained operators, and it cannot be used for an early diagnosis. It is also likely to miss cases with low infection intensities, offers poor sensitivity, and thus has a high rate of false-negative results in low prevalence regions [4,5,6,7,8]. Immunological tests are widely used in endemic areas, and the usefulness of some immunological tools has been demonstrated, particularly for symptomatic clinical cases. However, they cross-react readily with other parasitic diseases, causing false-positive results, and cannot distinguish past infections from current infections [6,9,10,11,12,13]. It is also difficult to diagnose schistosomiasis japonica in different animal species because the commercial secondary antibodies used in immunological tests are limited, and *S. japonicum* can infect more than 40 kinds of mammals [10,14]. As an alternative, point-of-care rapid test for the detection of *Schistosoma* circulating cathodic antigen (POC-CCA) in urine and serum samples is available. However, the performance was dissatisfactory in low endemic areas. This method does not work well in *S. haematobium* detection and it gives false-positive results [3,6,15,16,17].

In China, with long-term large-scale chemotherapy campaigns, the infection rates and infection intensities in epidemic areas have been maintained at relatively low levels in recent years. Therefore, the currently available diagnostic methods are inadequate for the accurate diagnosis of schistosomiasis in low-prevalence and low-intensity infection areas. Furthermore, the lack of an effective screening method for animals and humans could easily lead to the disease existing continuously in areas with ecological and environmental conditions that satisfy the life cycle of the parasite [18]. In order to design more optimal schistosomiasis control programs, upgraded diagnostic methods have great prospects to gain baseline information on the true prevalence. Meanwhile, precise diagnosis is important for monitoring the control measures and for patient management [3,19].

Molecular diagnostic technologies, especially nucleic acid detection, have opened a new chapter in the diagnosis of schistosomiasis. In recent years, with the development of molecular biology and genomics and the availability of genomic data for *Schistosoma*, various molecular diagnostic methods, including PCR, have been widely used to detect schistosome nucleic acids [20,21,22]. These methods have demonstrated greater specificity and sensitivity and less cross-reactivity than traditional methods [10]. Several schistosome target sequences (e.g., 5D, SjR2, and 18S rRNA) and some mitochondrial genes have been screened and verified for the diagnosis of schistosomiasis in previous studies. An SjR2-pCR2.1 recombinant plasmid template was detected with loop-mediated isothermal amplification (LAMP) at a limit of detection of 10^−4^ ng, and LAMP was more sensitive than conventional PCR [4]. Several molecular diagnostic methods have also been tested to evaluate their potential utility in the early diagnosis and evaluation of therapies for schistosomiasis. A 230-bp DNA fragment of the *S. japonicum* genome was detected in rabbit sera in the first week post-infection, and was no longer detectable 10 weeks after treatment [5]. The LAMP method was established based on the sequence of the highly repetitive retrotransposon SjR2, which detected *Schistosoma* DNA in the sera of rabbits at 1-week post-infection. The sensitivity of LAMP was 96.7% when testing 30 serum samples from *S. japonicum*-infected patients. After praziquantel (PZQ) treatment, no *S. japonicum* target DNA was detected in rabbit sera at 12 weeks after treatment [23]. A nested PCR (nPCR) method using SjR2 as the target gene was also established for the diagnosis of schistosomiasis japonica in livestock, with positivity rates of 92.30% (36/39) and 100% (39/39) at 14 dpi and 28 dpi, respectively [24].

Schistosomiasis japonica is gradually being eliminated in many endemic areas in China, and the infection rate and infection intensity of this disease decrease from year to year. Therefore, the goal of the national program has shifted from the control of schistosomiasis to its elimination. Consequently, diagnostic methods with a greater sensitivity and specificity are urgently required to detect this disease precisely. In this study, a qPCR method as well as a recombinase polymerase amplification (RPA) and lateral-flow dipstick (LFD) assay reported in our previous studies were evaluated for their diagnostic efficacy in the detection of early infections of *S. japonicum* and infections of different intensities.

## 2. Materials and Methods

### 2.1. Parasites and Animals

*Schistosoma japonicum* was maintained in the Shanghai Veterinary Research Institute. Specific pathogen-free BALB/c mice (4–6-weeks-old) were percutaneously infected with different amounts of *S. japonicum* cercariae for 15 min after the cercariae were counted under a microscope. Adult schistosomes were collected from the infected mice at 40 dpi. After the abdominal wool of eight goats was shaved, they were percutaneously infected with 300 *S. japonicum* cercariae for 20 min. Schistosomes were then mainly collected through the hepatic portal vein by perfusion with phosphate-buffered saline (PBS) containing 1% sodium citrate. The animals were carefully checked for schistosomes, which were manually removed from the mesenteric veins, and the number of worms in each mouse or goat were counted. All of the procedures performed on animals in this study were conducted following institutional ethical guidelines that were approved by the Animal Care and Use Committee of Shanghai Veterinary Research Institute, Chinese Academy of Agricultural Sciences (SHVRI-SZ-20200218-01).

### 2.2. Sample Collection

Blood samples were collected from the retro-orbital vessels of the mice into both EDTA-K2 vacuum blood collection tubes and Eppendorf tubes. Blood was collected from the jugular vein of each goat into both EDTA-K2 vacuum blood collection tubes and vacuum blood collection tubes with no additive. Samples collected from several mice and goats before infection were used as the corresponding negative controls.

After the EDTA-K2 vacuum tubes were centrifuged at 1000× *g* for 10 min at 25 °C, the plasma samples were isolated as the supernatants. The serum samples were collected after the Eppendorf tubes or vacuum blood collection tubes with no additive were centrifuged at 1000× *g* for 10 min at 25 °C.

### 2.3. DNA Extraction

Cell-free DNA was extracted from the plasma samples (0.1 mL/mouse sample; 0.6 mL/goat sample) with a Magnetic Serum/Plasma DNA Maxi Kit (Tiangen Biotech, Beijing, China). Worm genomic DNA was extracted with the TIANamp Genomic DNA Kit (Tiangen Biotech), according to the manufacturer’s protocol. All of the extracted DNA samples were stored at −20 °C.

### 2.4. The qPCR Assay for the Diagnosis of Schistosomiasis Japonica

The qPCR assay developed in our previous study [10] was performed with a 20 μL reaction mixture containing 10 μL of 2 × ChamQ Universal SYBR^®^ qPCR Master Mix (Vazyme, Nanjing, China), 0.4 μL (10 μmol/L) of forward primer, 0.4 μL (10 μmol/L) of reverse primer, 4 μL of extracted DNA template, and 5.2 μL of ddH_2_O. In each assay, the no template control contained ddH_2_O instead of the DNA template; in the negative control, DNA extracted from mice or goats before infection was used as the template; and in the positive control, *S. japonicum* DNA was used as the template. The PCR cycling conditions were 3 min denaturation at 94 °C, followed by 40 cycles of 15 s denaturation at 94 °C, 34 s annealing at 58 °C, and 10 s extension at 72 °C. The results were determined (as the characteristic peaks) with a melting curve analysis.

### 2.5. Recombinase Polymerase Amplification (RPA) Combined with a Lateral-Flow Dipstick (-LFD Assay)

The RPA–LFD assay was developed and optimized in our previous study [25]. In brief, 50 μL of reaction mixture contained 2.1 μL (10 μM) each of the forward primer and reverse primer, 0.6 μL (10 μM) of probe, 29.5 μL of primer-free rehydration buffer, 12.2 μL of nuclease-free water, and 1 μL of DNA template. Finally, 2.5 μL (280 mM) of magnesium acetate (MgOAc) was added, and reaction mixture was mixed to start the reaction. Amplification was conducted at 39 °C for 15 min, and 5 μL of the product was diluted to 100 μL with a running buffer for the LFD assay. The result was adjudged according to the band produced after 5 min at room temperature.

### 2.6. qPCR and RPA–LFD Detection of S. japonicum in Mice at Different Times after Infection and with Different Infection Intensities

Plasma samples were collected from 27 artificially infected mice at different infection times. These mice were infected with 5, 10, or 40 cercariae, and were killed at 40 dpi. Plasma samples were collected every day in the first week, and at 10, 14, 20, 30, and 40 dpi. Cell-free DNA was extracted from these samples, and analyzed with both qPCR and LFD–RPA.

### 2.7. Detection of S. japonicum in Goats with qPCR and RPA–LFD at Different Times after Infection

The plasma samples were collected from eight artificially infected goats at different times after infection. These goats were infected with 300 cercariae and the plasma samples were collected every day in the first week post-infection, and at 14, 24, and 56 dpi. Cell-free DNA was extracted from all of the plasma samples and analyzed with qPCR and RPA–LFD.

### 2.8. Data Analysis

The 95% confidence intervals (CI) for the specificity and sensitivity of the assays were calculated with Stata/SE 12.0 (College Station, TX, USA).

## 3. Results

### 3.1. Detection Efficacy of qPCR in Mice with Different Infection Intensities

A total of 27 plasma samples from mice infected with different numbers of cercariae were isolated and analyzed with qPCR. The results showed that 24 samples were positive at 40 dpi, so the total sensitivity of the qPCR assay was 88.9% (24/27, 95% CI: 70.84–97.65%). None of the 10 negative control samples tested positive, so the specificity of the qPCR assay was 100% (10/10, 95% CI: 69.15–100%).

The positive detection rates were calculated according to the different numbers of cercariae used to infect the mice (Table 1). In mice infected with five cercariae, the positive detection rate was only 77.8% (7/9, 95% CI: 39.99–97.19%) at 40 dpi, whereas when the mice were infected with 10 cercariae, the positive detection rate was 90% (9/10, 95% CI: 55.50–99.75%) at 40 dpi, and this increased to 100% (8/8, 95% CI: 63.06–100%) in the mice infected with 40 cercariae.

In three mice that were negative for qPCR assay, one pair of worms, one pair plus one male worm, and one pair plus another three male worms were detected (Table 2). Overall, the qPCR assay was highly sensitive at 40 dpi when the mice were moderately or severely infected with schistosomes, but was less sensitive at 40 dpi for low-intensity infections in mice.

### 3.2. Detection Efficacy of qPCR for Schistosomiasis Japonica at Different Stages of Infection

The qPCR results showed that the positive detection rate was ≤50% in the first week post-infection if the mice were infected with less than 10 cercariae, and detection at 3–4 dpi was dependent on the number of cercariae with which the mice were initially infected. The positivity rate then decreased and fluctuated, and was usually lowest between 10 and 20 dpi. However, the positive detection rate in all of the mice subjected to three different infection intensities was highest at 40 dpi. When the mice were infected with 40 cercariae, the positive detection rate was almost 90% after 30 dpi (Table 1).

In goats infected with schistosomes, the trend in the positive detection rate with qPCR was similar to that in mice (Table 3). *Schistosoma japonicum* was first detected at 3 dpi, and then the rate of positivity decreased to a low level, before ultimately peaking at 100% at 56 dpi. The worm burden of each infected goat was from 6 to 132 (Table 4).

### 3.3. Detection of S. japonicum with RPA–LFD Assay in Mice with Different Infection Intensities

Twenty-one of the 27 samples were positive for *S. japonicum* at 40 dpi, so the sensitivity of the RPA assay was 77. 8% (21/27, 95% CI: 57.74–91.38%) (Table 5). None of the 10 negative control samples tested positive, so the specificity of the RPA–LFD assay was 100% (10/10, 95% CI: 69.15–100%).

The positive detection rates were calculated according to the different numbers of cercariae used to infect the mice (Table 5). When the mice were infected with five cercariae, the positive detection rate was only 55.6% (5/9, 95% CI: 21.20–86.30%) at 40 dpi. When the mice were infected with 10 cercariae, the positive detection rate was 80% (8/10, 95% CI: 44.39–97.48%) at 40 dpi, but reached 100% (8/8, 95% CI: 63.06–100%) when the mice were infected with 40 cercariae.

### 3.4. Detection of Schistosomiasis Japonica at Different Stages with the RPA–LFD Assay

With RPA–LFD, ≤40% of infected mice were positive for *S. japonicum* in the first week post-infection, although the mice infected with 10 cercariae tested positive at 4 dpi. The positive detection rate in week 1 after infection peaked at 4–5 dpi, which was 1 day later than the peak of detection with qPCR. The positive rate then decreased to 0 between 10 and 14 dpi, and increased again after 20 dpi. The positive detection rate in the mice subjected to all three different infection intensities was the highest at 40 dpi. When the mice were infected with five cercariae, the positive detection rate was only 55.6% at 40 dpi, but was 100% when the mice were infected with 40 cercariae (Table 5).

In goats, the trend in the positive detection rate with RPA–LFD differed slightly from that in mice (Table 3). The positive detection rate was 37.5% at 1 dpi, peaked at 3–4 dpi (25.0%), and then decreased to a low level (≤12.5%). However, it was 100% at 56 dpi.

## 4. Discussion

Throughout nearly 70 years of extensive efforts to prevent and control schistosomiasis japonica, including large-scale treatment with PZQ, safety education, and extensive comprehensive prevention and control strategies, the morbidity and transmission of this disease have been maintained at very low levels in China. Today, the elimination of schistosomiasis japonica has been placed on the agenda of the national strategic plan of Healthy China 2030. However, some risk factors for disease transmission still exist in regions of low prevalence and low-intensity infections. Therefore, extremely sensitive and specific detection methods are urgently required to detect cases of the disease and monitor its epidemic status [26].

Rapid and reliable diagnostic techniques that can accurately identify the target groups for treatment are central to the prevention and control of schistosomiasis [27]. In previous studies, many nucleic acid detection methods, including general PCR, nPCR, qPCR, digital PCR, RPA–LFD, and so on, have demonstrated greater sensitivity and specificity and less cross-reactivity than traditional methods [10,28,29,30,31]. DNA-based diagnostic techniques have also been used to detect schistosomal DNA in early infections and after PZQ treatment. One study showed that LAMP and nPCR methods have potential utility in the detection of early infections in rabbits [4].

In the present study, a qPCR diagnostic method reported in our previous study was used to detect schistosomiasis japonica in mice. This method has shown greater sensitivity and specificity than soluble egg antigen (SEA)–ELISA in goats, and no cross-reactivity with eight other parasite species (*Haemonchus contortus*, *Fasciola gigantica*, *Toxoplasma gondii*, *Sarcocystis* sp., *Trichinella spiralis*, *Paramphistomum*, *Babesia*, and *Spirometra mansoni*) [10]. In mice, the method also showed a high sensitivity and specificity (unpublished). These results were largely consistent with the diagnostic results for goat schistosomiasis. According to the result of the three mice in which *S. japonicum* was not detected, if only a small number of paired worms survived in the host, the schistosomal nucleic acids from the worms and eggs would be very low, which may have directly caused the low concentration of schistosomal nucleic acids in the host blood. Therefore, a very small number of worms or paired worms may be a reason for a false-negative diagnosis. When the number of worms in the mice exceeded 10, the positive detection rate ultimately reached 100%.

The mice infected with the three different amounts of cercariae were tested with qPCR. The first peak of positive detection appeared at 3–4 dpi, and the rate of detection then decreased. This peak may indicate the time at which the nucleic acids of *S. japonicum* were released into the host blood circulatory system from the dead bodies of schistosomula during the process of invasion and migration to the lung. As the nucleic acids released by the schistosomula were degraded or metabolized by the mice, the rate of positive detection declined. Then, the schistosomula might gradually become suited to the environment of hosts and survive well, so the positive rate of detection remained at a low level. After 28 dpi, the positive detection rate increased again, and reached almost 100% at 40 dpi. During this stage, the female worms spawned many eggs per day and the schistosomal nucleic acids may have been predominantly derived from the disintegration of these eggs. These results are largely consistent with a study of the metabolic dynamics of *S. japonicum* DNA in the sera of rabbits [32]. The qPCR analysis of schistosome-infected goats showed a similar trend in the positive detection rate during the infection process. Therefore, 3 or 4 dpi would be the best detection point in the first week post-infection when qPCR is used for the diagnosis of infection, although not all of the infected samples could be completely checked out.

In a previous study, a PCR assay detected *S. japonicum* DNA in rabbit sera 1-week post-infection when the rabbits were infected with different numbers of cercariae [5]. The LAMP method also detected specific *S. japonicum* DNA in rabbit sera at 1-week post-infection [23]. Wang et al. used PCR and LAMP methods to detect *S. japonicum* in infected rabbits. The LAMP assay detected schistosomal DNA in all of the blood samples from infected rabbits at 1 week post-infection, but the PCR method did not detect it in all the samples [4]. The discrepancies in these results may be attributable to the different DNA extraction methods, the different detection methods, and the different target sequences used. To understand the metabolic dynamics of *S. japonicum* DNA in the serum of its host, an nPCR assay was used to analyze the sera of the rabbits infected with monosexual or mixed sexual cercariae as early as 3 dpi, and all of the samples were positive [32]. Another nPCR assay was used to detect specific *S. japonicum* DNA in the sera of mice at 3, 5, 7, and 14 dpi after infection with 5, 10, or 20 cercariae. The positive rate ranged from 80% to 90% at four detection points in mice treated with the three doses of cercariae [33]. In the present study, the detection rate of the qPCR assay in the first week post-infection did not reach that of the previously reported nPCR methods. The reason might be that nPCR requires two rounds of PCR, whereas qPCR involves only one round. However, the qPCR assay has the advantage of a lower risk of contamination than nPCR, especially in the detection of a large numbers of samples.

We also used an RPA–LFD diagnostic method reported in our previous study to detect schistosomiasis japonica in mice and goats at different stages of infection. Overall, the sensitivity of RPA–LFD was lower than that of qPCR in mice, especially in the first 3 dpi and at 40 dpi with low or moderate intensity infections. This might be because the qPCR method was more sensitive than the RPA–LFD assay. The qPCR detected 0.26 fg of target DNA, whereas RPA–LFD only detected 2.6 fg of target DNA [25]. Although RPA–LFD showed a high sensitivity and specificity in the diagnosis of schistosomiasis japonica in previous studies [25,28], it could be the most suitable for the diagnosis of moderate and severe infections after 40 dpi. The positive detection rate reached 88.9% (16/18) in mice and 100% in goats (8/8). qPCR may be slightly more accurate than RPA–LFD in the diagnosis of schistosomiasis japonica after the first week of infection. However, it was puzzling that RPA–LFD had greater diagnostic efficacy than qPCR at 20 dpi in mice. Nonetheless, RPA–LFD is an alternative and potentially useful method for the diagnosis of moderate and severe infections of *S. japonicum* after 40 dpi because it is simple, rapid, and has no requirement for advanced instrumentation.

Molecular diagnoses have significant advantages in their sensitivity and specificity, and thus have potential utility in the early diagnosis of *S. japonicum* infection. In the present study, the qPCR and RPA–LFD assays were used to continuously monitor the content and change the target sequence, especial in the first week. This helped us to understand the real nucleic acid metabolism of schistosome in the final hosts after cercariae infection. The results also revealed that not all highly repetitive sequences were suitable for the early diagnosis of *S. japonicum* infection, although they might exhibit a good diagnosis effect for schistosomiasis japonica. Further investigations are required to identify more target sequences for the early diagnosis of *S. japonicum* infection. The real worm burden was an important indicator. The actual number of viable and infective schistosome may be different despite hosts infected with the same number of cercariae. Sometimes, the same infection amount would cause a false assessment of the sensitivity and accuracy for diagnosis. In addition, the egg number in host might be another critical factor, and we would add this analysis into further diagnosis studies for schisotosomiasis.

The sensitivity of the molecular diagnostic methods means that aerosol contamination from the environment when PCR tubes are opened can easily cause misdiagnosis. Therefore, other molecular detection methods that do not require a tube to be opened, and in which the results can be observed directly, should be investigated. In addition, sample preparation and DNA extraction are still the key factors that limit the use of molecular diagnostics in POCT and some resource-poor areas due to the equipment requirements, time needs, and costs [34]. Some simple, fast, efficient, and few-equipment requirement extraction methods are urgently needed and there may be a tendency towards molecular diagnostics used in POCT in the future.

In conclusion, although the qPCR and RPA–LFD assays were not remarkable in the early diagnosis of *S. japonicum* infection, they are still useful and alternative molecular methods for the detection of schistosomiasis japonica in both mice and goats. Further investigations are required to identify more target sequences for early diagnosis, as well as the combination of different techniques with suitable target sequences.

## Figures and Tables

**Table 1 microorganisms-11-01059-t001:** Detection efficacy of the qPCR assay for mouse schistosomiasis japonica at different stages.

Time (p.i.)	Infected with Five Cercariae	Infected with Ten Cercariae	Infected with 40 Cercariae
	Positive Rate (No. of Positive Samples/No. of Total Samples)	Positive Rate (No. of Positive Samples/No. of Total Samples)	Positive Rate (No. of Positive Samples/No. of Total Samples)
1 d	22.2% (2/9)	20.0% (2/10)	25.0% (2/8)
2 d	22.2% (2/9)	20.0% (2/10)	50.0% (4/8)
3 d	44.4% (4/9)	40.0% (4/10)	50.0% (4/8)
4 d	22.2% (2/9)	50.0% (5/10)	75.0% (6/8)
5 d	22.2% (2/9)	30.0% (3/10)	37.5% (3/8)
6 d	33.3% (3/9)	30.0% (3/10)	50.0% (4/8)
7 d	44.4% (4/9)	20.0% (2/10)	50.0% (4/8)
10 d	0% (0/9)	0% (0/10)	12.5% (1/8)
14 d	0% (0/9)	0% (0/10)	12.5% (1/8)
20 d	0% (0/9)	0% (0/10)	25.0% (2/8)
30 d	33.3% (3/9)	40.0% (4/10)	87.5% (7/8)
40 d	77.8% (7/9)	90.0% (9/10)	100.0% (8/8)

**Table 2 microorganisms-11-01059-t002:** The worm burden of each mouse infected with different amounts of cercariae.

Mouse No.	Infected with Five Cercariae			Infected with Ten Cercariae			Infected with 40 Cercariae		
	Pair	Single	Total	Pair	Single	Total	Pair	Single	Total
1	1	1	3	3	1	7	10	1	21
2	2	0	4	3	1	7	5	3	13
3	1	1	3	1	3	5	8	3	19
4	0	2	2	1	2	4	4	3	11
5	1	0	2	1	2	4	6	1	13
6	2	0	4	3	2	8	9	2	20
7	1	0	2	2	0	4	10	1	21
8	1	0	2	3	0	6	17	3	37
9	2	0	4	2	2	6	\	\	\
10	\	\	\	2	0	4	\	\	\

The gray background means that the mouse was not checked out by the qPCR.

**Table 3 microorganisms-11-01059-t003:** Detection efficacy of the qPCR and RPA–LFD assay for goat schistosomiasis japonica at different stages.

Time (p.i.)	qPCR	RPA–LFD
	Positive Rate (No. of Positive Samples/No. of Total Samples)	Positive Rate (No. of Positive Samples/No. of Total Samples)
1 d	12.5% (1/8)	37.5% (3/8)
2 d	25.0% (2/8)	0% (0/8)
3 d	50.0% (4/8)	25.0% (2/8)
4 d	12.5% (1/8)	25.0% (2/8)
5 d	12.5% (1/8)	0% (0/8)
6 d	0% (0/8)	0% (0/8)
7 d	0% (0/8)	12.5% (1/8)
14 d	12.5% (1/8)	0% (0/8)
24 d	12.5% (1/8)	0% (0/8)
56 d	100.0% (8/8)	100.0% (8/8)

**Table 4 microorganisms-11-01059-t004:** The worm burden of each infected goat.

Goat No.	Pair	Single	Total
1	27	11	65
2	2	2	6
3	2	2	6
4	11	0	24
5	8	12	28
6	6	1	13
7	4	9	17
8	31	70	132

**Table 5 microorganisms-11-01059-t005:** Detection efficacy of the RPA–LFD assay for mouse schistosomiasis japonica at different stages.

Time (p.i.)	Infected with Five Cercariae	Infected with Ten Cercariae	Infected with 40 Cercariae
	Positive Rate (No. of Positive Samples/No. of Total Samples)	Positive Rate (No. of Positive Samples/No. of Total Samples)	Positive Rate (No. of Positive Samples/No. of Total Samples)
1 d	0% (0/9)	0% (0/10)	0% (0/8)
2 d	0% (0/9)	10.0% (1/10)	0% (0/8)
3 d	0% (0/9)	10.0% (1/10)	0% (0/8)
4 d	22.2% (2/9)	60.0% (6/10)	25.0% (2/8)
5 d	22.2% (2/9)	30.0% (3/10)	37.5% (3/8)
6 d	22.2% (2/9)	10.0% (1/10)	12.5% (1/8)
7 d	0% (0/9)	10.0% (1/10)	0% (0/8)
10 d	0% (0/9)	0% (0/10)	0% (0/8)
14 d	0% (0/9)	0% (0/10)	0% (0/8)
20 d	33.3% (3/9)	40.0% (4/10)	37.5% (3/8)
30 d	33.3% (3/9)	50.0% (5/10)	87.5% (7/8)
40 d	55.6% (5/9)	80.0% (8/10)	100.0% (8/8)

## Data Availability

Not applicable.

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
