# Peer review of "Two Molecular Plasma-Based Diagnostic Methods to Evaluate Early Infection of Schistosoma japonicum and Schistosomiasis Japonica"

_microorganisms, 2023, doi:10.3390/microorganisms11041059_

Round 1

Reviewer 1 Report

This manuscript describes the assessment of two methods for the diagnosis of S. japonicum. Using experimental infections of mice and goats, the efficacy and sensitivity of a molecular PCR based assay was compared to a lateral-flow dipstick. By using different infection doses, the authors could compare the sensitivity of both diagnostic tests, revealing no significant difference. For the detection of early infection, a critical need for accurate surveillance and control, neither assay proved to be particularly useful with positivity rates of approx. 30% in both animals at 1 dpi.  

Standard approaches were used for the comparison of samples and appeared to have been performed and analysed appropriately.

My major comment is that the numbers of worms retrieved from the infected animals is actually a critical piece of data and would be better embedded into the tables of the main manuscript rather than in supplementary data. Even though mice/goats may have been infected with x number of parasites – the actual number of viable and infective parasites may well be different, thus giving a false assessment of sensitivity and accuracy. While this aspect of the study is covered in the discussion, I feel it would be a better fit within the results section as it is a critical piece of information which is needed to more accurately determine the efficacy of the diagnostic assays.

The Discussion is a little lengthy for the amount of data and results that are presented and could be shortened slightly.

Author Response

My major comment is that the numbers of worms retrieved from the infected animals is actually a critical piece of data and would be better embedded into the tables of the main manuscript rather than in supplementary data. Even though mice/goats may have been infected with x number of parasites – the actual number of viable and infective parasites may well be different, thus giving a false assessment of sensitivity and accuracy. While this aspect of the study is covered in the discussion, I feel it would be a better fit within the results section as it is a critical piece of information which is needed to more accurately determine the efficacy of the diagnostic assays. 

Response: Thanks for the kindly suggestion. We have added these data in the main body of revised manuscript. (Table 2 and Table 4)

The Discussion is a little lengthy for the amount of data and results that are presented and could be shortened slightly.

Response: Thanks for the comment. According to the word number requirement of the journal and another reviewer’s comment, we slightly shorten the Discussion and add a section to discuss strengths and weaknesses of our research.

Reviewer 2 Report

Thank you for sharing your manuscript on plasma-based diagnosis to detect schistosomiasis infection early. Here some comments and suggested edits that could help to improve the article:

L52-53: Please briefly outline in your manuscript which parasitological and immunological diagnostic assays within the national schistosomiasis control program in China you are referring to.

L59: Please correct "require" to "requires".

L61: Please correct "offer" to "offers".

L67-68: Using "commercial" and "commercially" in the same sentence seems duplicated; please revise.

L70: Should it be "are available" or "is available"?

L75/76: Please specify the time point(s) "now" and "current situation".

L155: "lateral" seems duplicated.

L271: Consider rephrasing "would very low" to "would be very low".

L265: Please name the other parasite species in your manuscript. 

L334-335: Please explain in more detail what you mean by aerosol contamination as nucleic acid-based methods should in a common understanding be prepared under controlled conditions such as a closed flow/hood in addition to using filtered equipment throughout.

Generally, please add a section to discuss strengths and weaknesses of your research as well as a conclusion section. 

Author Response

L52-53: Please briefly outline in your manuscript which parasitological and immunological diagnostic assays within the national schistosomiasis control program in China you are referring to.

Response: Thanks for the comments. We have added the corresponding diagnostic assay methods in the revised manuscript. (line 53-55)

L59: Please correct "require" to "requires".

Response: Thanks for the suggestion. We have revised the word. (line 61)

L61: Please correct "offer" to "offers".

Response: Thanks for the suggestion. We have revised the word. (line 62)

L67-68: Using "commercial" and "commercially" in the same sentence seems duplicated; please revise.

Response: Thanks for the suggestion. We have deleted the ‘commercially’ in line 69.

L70: Should it be "are available" or "is available"?

Response: Thanks for the comment. We have changed it into ‘is available’. (line 72)

L75/76: Please specify the time point(s) "now" and "current situation".

Response: Thanks for the suggestion. We have changed ‘until now’ into ‘in recent years’, and changed ‘in current situation’ into ‘in low-prevalence and low-intensity infection areas’. (line 77-79)

L155: "lateral" seems duplicated.

Response: Thanks for the comment. We are sorry for the mistake and we have deleted one ‘lateral’. (line 157)

L271: Consider rephrasing "would very low" to "would be very low".

Response: Thanks for the comment. We have changed it to ‘would be very low’. (line 274)

L265: Please name the other parasite species in your manuscript. 

Response: Thank you for the comment. We have added the names of the eight parasite species in revised manuscript. (line 268-269)

L334-335: Please explain in more detail what you mean by aerosol contamination as nucleic acid-based methods should in a common understanding be prepared under controlled conditions such as a closed flow/hood in addition to using filtered equipment throughout.

Response: Thanks for the comment. In an orthonormal detection lab, all the conditions and factors are under control. Just like what reviewer’s mentioned, all the operations could be finished in different rooms, which were used filtered equipment throughout. In this condition, aerosol contamination is not easy to happen. However, nucleic acid-based methods are used in lab without filtered equipment or a closed flow/hood sometimes, and they may also be operated in POCT. Moreover, some detections will use the amplified products of PCR or RPA for further analysis, such as electrophoresis.

Generally, please add a section to discuss strengths and weaknesses of your research as well as a conclusion section. 

Response: Thanks for the suggestion. We have added the section in the revised manuscript. (line 345-348)

Round 2

Reviewer 2 Report

Thank you for sharing the revised manuscript. Overall, most of my comments were addressed sufficiently. 

Author Response

We are sorry that we omit the response of the last comment. We have added and revised the section in line 328-341 and 351-355 according to reviewer’s and academic editor’s suggestion.